# The Role of Twitter in the WHO’s Fight against the Infodemic

**DOI:** 10.3390/ijerph182211990

**Published:** 2021-11-15

**Authors:** Daniel Muñoz-Sastre, Luis Rodrigo-Martín, Isabel Rodrigo-Martín

**Affiliations:** Campus María Zambrano, Universidad de Valladolid, 40005 Segovia, Spain; daniel.munoz.sastre@uva.es (D.M.-S.); luis.rodrigo@uva.es (L.R.-M.)

**Keywords:** infodemic, COVID-19, vaccination, Twitter, World Health Organization

## Abstract

The COVID-19 pandemic has far-reaching consequences in various fields. In addition to its health and economic impact, there are also social, cultural and informational impacts. Regarding the latter, the World Health Organization (WHO) flagged concerns about the infodemic at the beginning of 2020. The main objective of this paper is to explore how the WHO uses its Twitter profile to inform the population on vaccines against the coronavirus, thus preventing or mitigating misleading or false information both in the media and on social networks. This study analyzed 849 vaccine-related tweets posted by the WHO on its Twitter account from 9 November 2020 (when the 73rd World Health Assembly resumed) to 14 March 2021 (three months after the start of vaccination). In order to understand the data collected, these results were compared with the actions carried out by the WHO and with the information and debates throughout this period. The analysis shows that the WHO is decidedly committed to the use of these tools as a means to disseminate messages that provide the population with accurate and scientific information, as well as to combat mis- and disinformation about the SARS-CoV-2 vaccination process.

## 1. Introduction

Communication is one of the essential tools in managing a health crisis such as the pandemic caused by the spread of SARS-CoV-2. Citizens increasingly demand more information in this regard. In this type of situation, the spokespersons of the institutions that manage these crises should give the population “the information the public needs and counter some of the harmful behaviors that are common during an emergency, so we can effectively support the public, our colleagues, and the organizations that are offering help”, as noted by the Centers for Disease Control and Prevention (CDC) of the United States [1].

As the demand for information increases, the amount of fake news also grows. In light of this situation, it is necessary to establish a protocol to combat the misinformation caused by such false news. The correction of disinformation is a topic addressed by authors such as Lewandowsky et al., who suggest a series of recommendations to develop tools to combat disinformation [2]. These authors also point to reasons that may contribute to the resistance to the correction of disinformation, such as the fragmentation of the information landscape by the new media or the creation of coherent stories [2].

Before proceeding further, in order to be able to approach the subject of the study with greater clarity, it is necessary to propose a differentiation between the terms “disinformation” and “misinformation”. Freelon and Wells [3] make a differentiation based on what they call “the cognitive domain.” Thus, it may be said that “disinformation” refers to the deliberate creation or sharing of false information, whereas “misinformation” is not intended to mislead the receiver.

The fight against disinformation is also addressed by Bode and Vraga [4] in an analysis of social networks as an element to combat false information in moments of crisis. These authors analyze the processes of disinformation during the spread of the Zika virus and propose important bases for the future study of the role of social media in the fight against fake news.

This paper is the result of our research on the use of social networks by public organizations. We not only focus on their communications to the public but also seek to “identify patterns and behaviors linked to the search and provision of health-related information” [5]. Since the beginning of the SARS-CoV-2 pandemic, social networks have become one of the main forums for debate on the various issues related to the global spread of COVID-19. However, the information published on these platforms often lacks scientific support, which adds to the confusion, mistrust and fear among the population.

Barely a month and a half after confirming the first cases of SARS-CoV-2 infection in China, in February 2020, the World Health Organization (WHO) warned of the existence of an infodemic around COVID-19. Director-General of the WHO, Tedros Adhanom, noted that “the evolution of the coronavirus outbreak will depend on the extent to which the correct information is delivered to the people who need it” [6].

The term infodemic did not appear with the outbreak of coronavirus, nor is it exclusive to this pandemic. In fact, the WHO has been using this term for some time. Additionally, some authors already explored the role of certain communication tools in the dissemination of misinformation on health-related issues (see Pulido et al. [7]).

However, this concept (which according to the Cambridge Dictionary refers to “a situation in which a lot of false information is being spread in a way that is harmful” [8]) is particularly prominent in the SARS-CoV-2 crisis, largely due to the new role occupied by social networks and other digital communication tools. The oversaturation of such information is an obstacle in the fight against COVID-19, as the WHO itself acknowledges in several reports.

Since the beginning of the pandemic, the WHO has warned of the risks posed by the infodemic. The WHO’s Director-General went as far as to say, during a meeting of foreign policy and security experts in Munich (Munich, Germany), that “we are not only fighting a ‘pandemic,’ we are fighting an ‘infodemic’” [9]. Upon the outbreak of coronavirus, Dr. Adhanom and the vice-president of Tencent Healthcare, Alex Ng, called attention to the WHO’s use of social platforms to deliver clear, reliable and necessary information so that the population could receive reliable knowledge of world events. They also stressed the help provided by some of these networks in trying to stop the spread of the disease [6].

The importance given by the WHO to the management of information on the SARS-CoV-2 pandemic became apparent during the World Health Assembly in May 2020. Member states were encouraged to combat misinformation surrounding the health crisis through the use of existing digital communication tools. The Assembly also “call[ed] on international organizations to address mis- and disinformation in the digital sphere, work to prevent harmful cyber activities undermining the health response and support the provision of science-based data to the public” [10]. This request, included in Resolution WHA73.1, extended to the media, social networking platforms, technology experts and social leaders, so that all of them could cooperate in preventing the spread of misinformation with no scientific basis, while respecting freedom of expression [10].

Among these digital communication tools, social networks are prevalent for their social relevance and presence. According to a report by DataReportal, their global average penetration rate is 53.6% (reaching 79% in Western and Northern Europe, and 74% in North America [11]). These data confirm the importance of social networks in today’s communication processes and turn these digital tools into powerful communication tools.

Although Twitter is not the social platform with the largest number of users (332.4 million users in the world [12]), it has become a major digital forum for debate. Hence, it is relevant for understanding many current events, including those related to health (see the research carried out by Paul and Dredze on health and Twitter [13]). Alternatively, this social network is also an important communication tool for institutions, as illustrated by several studies (Burton et al. [14], Khan et al. [15] or Leone et al. [16] among others). Twitter’s power as a tool for institutional communication is not unidirectional; it allows institutions to connect with society [17], track public opinion and facilitate two-way communication [18].

The WHO posted its first tweet regarding the new coronavirus on 4 January 2020. Since then, the WHO has intensely engaged in dissemination efforts through Twitter, seeking to provide access to actual information both for the media and to the general public. Therefore, the WHO’s profiles across several social networks (particularly Twitter) became the organization’s loudspeakers to deliver accurate and reliable information, thus counteracting the negative effects of the infodemic that was spread in parallel with SARS-CoV-2.

Despite the WHO’s efforts to combat this information saturation and the over-dimensioning of fake news, several pieces of untrue or unfounded information have appeared since the outbreak of the pandemic. This unfounded information aggravated the situation by causing confusion and uncertainty, thus leading to erroneous opinions. In some instances, misinformation came from reliable sources, such as when the French Health Minister, Olivier Véran, published a tweet (Figure 1) warning of the risks posed by anti-inflammatories for COVID-19 patients [19]. However, this message, which had more than 39,400 retweets and 37,500 likes, was not technically considered fake news by some experts because it came from a reliable source [20].

The use of social networks as sources of information during the pandemic aroused great interest among communication experts and researchers. Particularly insightful is the analysis carried out by Nguyen and Catalán-Matamoros on the risks of disinformation, with examples ranging from anti-5G movements to anti-vaccine activists [21]. Aleixandre-Benavent, Castello-Cogollos and Valderrama-Zurián also explored the information and disinformation at the outset of the pandemic [22]. Other authors such as Depoux et al. focused on the use of social networks as tools to inform the public, and the contribution of these social networks in the fight against the virus. These authors suggested the need to consider more carefully the discussions that take place on these platforms in order to identify the source, track the information, and devise communication plans [23].

Regarding Twitter specifically, it is worth mentioning the work of Chen et al., which helped to identify false information published on Twitter regarding COVID-19 [24], as well as that of Kouzy et al., who, in the first months of the pandemic, already warned of the dizzying rate of the spread of fake news across social networks [25]. Rosenberg, Syed and Rezaie consider the advantages and disadvantages of Twitter for disseminating information, whether true or false, during the COVID-19 pandemic [26].

Research on the role of social networks in disseminating information (whether true or false) regarding the pandemic includes, notably, works that focus on specific territories, such as those of Han et al. on the Chinese case [27], or the work of Caliandro et al. in Italy [28]. Some studies analyze the individual profiles linked to public bodies or world leaders (see Rufai et al. [29]), while others assess the specific aspects of the pandemic, such as the origin of the virus (Budhwani and Sun [30]); the positioning of the media when it comes to covering COVID-19 [31] or vaccine-related issues (Catalán-Matamoros and Elías [32]). Together, this shows the great interest of the scientific community in everything related to pandemic communication, as well as the wide range of approaches to this type of research.

This paper presents the results of our research on the WHO’s use of Twitter to inform the public on the benefits of vaccination against COVID-19 in an attempt to combat mis- and disinformation on this subject. The main innovation of this research is the approach taken when analyzing the content of messages posted by WHO on its Twitter account about vaccines. The other contributions of this research are related to the criteria used in the selection of the sample and the procedure carried out for its analysis, which allow us to address the problem of the infodemic in the context of vaccines against COVID-19, based on an original source.

The reason why the research focused on vaccines was due to the interest that they generated in internet discussions. This discussion was not exclusive to the vaccine against COVID-19 but was generated on other occasions with other vaccines. This confrontation of positions was analyzed by authors such as Kata [33], who addressed the need to recognize the “post-modern discourses” defended by anti-vaccine activists, and thus opened a dialogue that resulted in the spread of misinformation. Another interesting approach to the use of social networks with regard to vaccines is the study carried out by Walter et al. [34] on the use of fake accounts by the Russian Internet Research Agency (IRA) to polarize the debate about vaccines in the USA.

Opposition to the vaccines increased when they were relatively new developments. The inexperience with these new vaccines increased the emergence of false beliefs that generated false information. The rejection of new vaccines is an issue addressed by authors such as Ophir and Jamieson [35] in their research on the Zika vaccine. This rejection was particularly evident with the introduction of vaccines against COVID-19. This issue was analyzed by different authors such as Romer and Jamieson [36], who focus the role of conspiracy theories in opposition to the vaccines against SARS-CoV-2.

The main purpose of our study is to understand how the WHO attempted to combat the infodemic surrounding SARS-CoV-2 vaccines through its Twitter account. Other secondary objectives are:To determine the nature of the messages;To understand the concepts most used in WHO’s tweets;To analyze the relation of these messages with potential hoaxes.

## 2. Materials and Methods

This study followed previous recommendations and guidelines from other authors, such as Zimmer and Proferes [37], for analyzing communication through social networks, which were implemented in other studies on the use of Twitter in institutional communication at the international level [38]. We applied these guidelines to our case, namely the WHO’s use of Twitter regarding the vaccinations against SARS-CoV-2.

For this purpose, we adopted a dual qualitative–quantitative approach. On the one hand, we analyzed the content of messages posted on the WHO’s account regarding COVID-19 vaccines. This consisted of identifying the terms most frequently used by this organization and determining their intent. This qualitative analysis yields quantifiable results (the number of occurrences of the most used terms or messages with a specific nature) to gather a representative picture.

On the other hand, this research is descriptive–analytical. In the first stage, we list the messages and the concepts used in them. In the second stage, we analyze the meaning of all these elements by searching for the possible links to events surrounding the vaccines and the vaccination process during the relevant period. In addition, this study has an explanatory value, since it seeks to determine the causes and consequences of our subject matter. To the extent that we consider the WHO’s tweets throughout a specific period of time, our study is longitudinal and retrospective.

The explicative nature of this research is based on the following initial hypothesis:

The World Health Organization uses its official Twitter account to publish messages that provide the public with accurate information on COVID-19 vaccines, and thus try to combat the existing misinformation and disinformation on this subject, adapting the contents of the topics that are at the focus of attention at any given time.

The authors asked themselves the following questions to lead the research:Which are the main themes that appear in the analyzed messages and what temporal evolution do they experience?Which are the main terms quoted in these WHO messages and how do they have a relative importance throughout the period under review?What possible intention may these messages have and how does this evolve during the weeks under review?Which are the relationships between the terms analyzed and possible WHO events or decisions regarding the SARS-CoV-2 pandemic?

Regarding the scope of the study, our research was based on a content analysis of the 699 tweets related to the SARS-CoV-2 vaccine posted by the WHO on its Twitter account [39] from 9 November 2020 (when the 73rd World Health Assembly resumed) to 14 March 2021 (three months after the start of vaccination). The initial date was established by taking into account the major WHO event closest to the approval of the first vaccines, while the closing date was set to coincide with the end of the third month since the start of vaccination. Hence, there is sufficient time to obtain results that reflect trends.

The variables proposed for the analysis are as follows:Number of messages posted by the WHO on Twitter regarding COVID-19 vaccines;Most frequently used concepts;Intent of the messages;Connection with topics in the news.

There are different formulas and tools for collecting data from Twitter, as explained by Mayr and Weller [40]. Given the objective and characteristics of this study, we opted for the use of third-party applications such as Twlets to collect messages [41], and More Than Books [42] and WordCounter.net [43] for word counts. Using the former app, on 15 March 2021, we gathered all the tweets posted by the WHO during the relevant period including the terms “vaccine,” “COVID-19” and/or “coronavirus.” Based on this data set, we applied a selection process to the resulting 3200 messages, using a conventional spreadsheet to identify and select those that referred to the SARS-CoV-2 vaccines. The objective of this second filter was to obtain a sample consisting of those messages directly linked to this vaccination process (with particular characteristics that made it different from other seemingly similar processes).

The second filter provided the final sample of 699 messages. We carried out a first analysis based on the most frequent terms, in order to identify the words most used by the WHO in its tweets about COVID-19 vaccines. This set the basis for the subsequent semantic analysis aimed at determining the nature of the messages after reviewing their content and following the prior studies by Colle [44] or Verd Pericás [45]. In this way, we obtained the key elements necessary to study the WHO’s behavior on its Twitter account during the relevant period. We drew up a timeline to identify the moments when messages about the vaccine were most numerous. Finally, we checked whether the increase in the number of tweets was related to the publication of fake and controversial news on this subject.

## 3. Results

### 3.1. The WHO and Its Fight against Infodemic

The WHO’s concerns regarding the risks of infodemic in the pandemic context led the organization to address this problem in parallel to the fight against the virus, as well as other social and health issues related to the SARS-CoV-2 crisis. This was best illustrated by the publication, in April 2020, of a report entitled ‘Managing the COVID-19 infodemic. Call for action’ [46]. This document brings together 50 recommendations from 1300 experts to tackle the coronavirus infodemic. Another example is the newly created section on the WHO’s website with true and accurate information that debunks false information on different aspects of the pandemic [47]. It also provides the following recommendations on how to avoid the infodemic [48]:Assess the source;Go beyond headlines;Identify the author;Check the date;Examine the supporting evidence;Check your biases;Turn to fact-checkers.

Furthermore, between 30 June and 16 July 2020, the WHO held the First WHO Infodemiology Conference [49], where experts from different fields discussed the possible measures to manage the infodemic. This was not the only meeting of this kind organized by the WHO, as similar meetings were held periodically to analyze the evolution of the infodemic and evaluate the effectiveness of the measures proposed, including new suggestions to improve its management. In parallel, the WHO held other meetings such as the EPI-WIN Webinars [50] on vaccine communication, among other issues.

On the other hand, the WHO is making a great effort to encourage citizens to report any false information they may receive through any channel, particularly virtual social platforms. For this purpose, it launched a campaign showing how to identify and report fake news on social networks [42]. This campaign is linked to the WHO’s collaboration with the UK Government under the slogan “Stop the Spread.” Its first phase focused on encouraging citizens to search reliable sources, while the second aimed to help identify and report hoaxes [51].

### 3.2. WHO’s Tweets on COVID-19 Vaccines

After the different selection processes based on the 9740 tweets posted by the WHO in the relevant period, the final sample consisted of 699 messages that referred directly to the subject of our study (i.e., COVID-19 vaccination) or were included in one of the threads on this issue. In order to analyze the messages, we classified them according to various criteria following models put forward by authors such as Strauss and Corbin [52], as shown in Table 1.

Bearing this in mind, it should be noted that most of the messages published by the WHO on its Twitter account during the 18 weeks under consideration are its own or original messages (89.4%). Of the total number of messages that were retweeted from other accounts, 60.8% corresponded to tweets published by WHO’s Director-General, Dr. Tedros and 74% of the messages in the final sample were directly related to our topic (i.e., they referred specifically to aspects linked to the SARS-CoV-2 vaccine). The remaining 26% were messages that did not make express reference to the vaccine but were included in threads that explained the benefits of vaccination or how to get a vaccine against a specific disease.

Our content analysis allowed us to determine the nature of the messages, giving priority to those with more than one intention. This showed that 58.8% of the 699 tweets analyzed had a purely informative content, while 18.3% were some kind of appeal by the WHO to other authorities, to specific groups, or to society as a whole. The rest of the messages made a warning (15.7%) or an announcement (7.2%), either of the approval of a vaccine or its shipment to a country.

This study also made it possible to define the topics most covered by these messages based on the terms they contained. Overall, we established seven thematic areas (see Figure 2 for the corresponding weight in the messages):Virus: including terms such as “COVID-19,” “SARS-CoV-2,” or “variants”;Vaccine: for messages related to the vaccine, either exclusively or through terms such as “safe,” “ensure,” “effective,” “approval,” “clot,” or the manufacturers;Solidarity: for concepts such as “vaccine equity,” “COVAX,” “nationalism,” “poor,” “together,” or “share”;Health workers: a collective often referred to by WHO’s tweets;Information/misinformation: a category that includes messages with references to terms such as “misinformation,” “mistrust,” or “information.”

Based on the density of terms that appear in the nearly 700 tweets analyzed, and taking into account the objective of the research, we selected a total of 23 words that appear regularly in these posts and were related to the object of study. These terms refer to the coronavirus that triggered the pandemic, to the disease caused by it, to the names of the vaccines, and to the issues related to the solidarity demanded by the WHO to ensure that vaccines could reach the most disadvantaged countries. The occurrence of these terms can be seen in Figure 3.

The next level of analysis allowed us to place the messages in a timeline to identify the predominant themes and concepts at each moment during the 18 weeks under consideration. Our first observation was that WHO’s activity on Twitter increased in the last four weeks, between 15 February and 14 March 2021, accounting for 42.2% of the messages analyzed. The week from February 22 to 28 registered the largest number of posts on the subject of our study (89 tweets).

However, such a higher volume of publications did not correspond to the highest number of interactions achieved in the seventh week of the study, between December 21 and 27, with 56.3% of favorites and 19.8% of retweets for all the messages in the sample.

Regarding the evolution of intent, the analysis shows that the purpose of the messages was mostly informative throughout the whole period, with the exception of the fifteenth week (15–21 February), when messages conveying different appeals took on a special prominence. As can be seen in Figure 4, most tweets belonged to one of these two categories, followed by “warnings”, most frequent during the twelfth and sixteenth weeks (i.e., 25–31 January and 22–28 February, respectively).

The individual evolution of the most frequent terms showed that “COVID-19” was the most mentioned in each of the weeks. The word “together” is the second most frequently used term in ten of the eighteen weeks, followed by “vaccine,” the second most frequently occurring word in six of the weeks studied. Figure 5 shows the number of weeks in which each term was among the three most frequently used.

However, if we group the terms by theme, solidarity predominates throughout the weeks under consideration, as can be seen in Figure 6. With the exception of three (out of eighteen) weeks, the terms related to solidarity are the most used in the rest of the period. The other two themes most present in the messages are those defined by terms associated with the name of the virus, its variants, or the disease caused by SARS-CoV-2, as well as everything related to the vaccines (whether direct references to the term “vaccine,” to security, or to manufacturers). Direct references to information/disinformation are a constant but minor presence, along with those to health workers.

Finally, the analysis attempts to establish a link between the prevalence of some terms and the events related to the evolution of the pandemic or WHO’s decisions. The predominance of references to “COVID-19” is self-explanatory; as the name of the disease caused by the virus that originated the pandemic, it should not be surprising that it appears in most of the messages referring to the subject of this research. The prominence of the word “together”, the second-most frequent in ten weeks, is linked to the WHO’s efforts to convey to governments in particular, and to society in general, the importance of acting in solidarity to ensure equal access to treatment for all. Indeed, this is the main purpose of the WHO’s campaign We Are #InThisTogether, launched on 10 November 2020 [53]. Something similar can be said of “vaccine,” the other second-most used term: its increasing occurrence in the last seven weeks of the study is related to the global process of vaccine distribution and inoculation.

## 4. Discussion

The research presented here allows us to reaffirm the positions previously held by other authors; that Twitter analysis is useful for extracting data on public health, as pointed out a decade ago by Paul and Dredze [13], and more recently by Gough et al. [54], Bode y Vraga [4], or by Gencoglu and Gruber [55]. Additionally, due to the nature and subject matter of a large part of the sample, this study also demonstrates the importance of social networks in disseminating the advances in public health referred to by Breland et al. [56]; in promoting vaccination, as noted by Javanainen [57]; and, ultimately, in saving lives, as pointed out by Nguyen and Catalán-Matamoros [21].

There is no doubt that social networks have a role beyond mere individual interaction. They connect citizens with their governments [17]. Given their growing prominence and impact, these digital tools should serve as transmission belts of truthful information between science and society. Health care managers and professionals should be aware of this fact and be prepared for it [5], and thus they should be acquainted with the dynamics of these types of tools in order to use them efficiently [14]. For this reason, it is very important to pay attention to the recommendations made by authors such as Lewandowsky et al. [2] on protocols to combat disinformation, and to the indications made by institutions such as the CDC [1].

On the other hand, the interest shown by the public in the WHO’s messages posted on Twitter is evidenced by the number of “likes” and retweets. This, in turn, confirms Twitter’s capacity to carry out communication campaigns to deliver accurate information in combatting disinformation caused by trolls and bots, as studied by Sutton [58]. The WHO’s active presence on Twitter is an important weapon against disinformation in this health crisis, as addressed, among others, by Pérez Dasilva et al. [59], Ramón Fernández [60], Kouzy et al. [25], and Rosenberg et al. [26].

Even if the messages analyzed do not include many direct references to terms such as “information,” “misinformation,” or “mistrust,” the WHO’s tweets comply with its own guidelines to combat false information [46,47,48,51]. In light of the WHO’s behavior and posts on Twitter, we can safely say that the SARS-CoV-2 pandemic signals a new threat in health crises: the infodemic. At the same time, the potential role of social networks, both in disseminating and combating fake news, seems undeniable [6,7,61].

With regard to the most frequent themes in the WHO’s tweets during the period analyzed, it is worth highlighting the prominence of solidarity in these eighteen weeks. This is linked to the WHO’s efforts since the beginning of the pandemic to fight the virus together, sharing knowledge and showing generosity to those who are most disadvantaged, whether individuals or countries. The WHO’s firm commitment to solidarity in managing the pandemic was addressed previously by Arora et al. [62], and it is evident in the results of this research.

The results obtained underline that the various institutions, who are competent in the field of health, should hold a debate in order to solve both the problem of misinformation arising from the refusal of vaccines [33,35,36] and the problems generated by interference from other countries [34].

## 5. Conclusions

The uncertainty created by the spread of SARS-CoV-2 makes it necessary to have reliable sources of information that reflect reality accurately, while promoting responsible attitudes among all social actors and reassuring the public. The World Health Organization demonstrated its awareness of this fact with its tweets on the pandemic and on coronavirus vaccines during the period analyzed.

The results showed that the choice of themes, the nature of the messages responding to changing needs and circumstances, and the publication of these tweets followed the dynamics of the pandemic. Hence, the WHO informed the public of the events and progress with science-based messages, while at the same time it made warnings, announcements and appeals to promote responsible behavior in order to face the pandemic together.

This study allows us to verify how the WHO’s narrative on its Twitter account bears witness to a period marked by significant advances in the fight not only against the pandemic, but also against the infodemic. The results demonstrate the regular use of the WHO’s Twitter account to disseminate information about vaccines against SARS-CoV-2 and to provide an overview of changes in tone and theme over the period under review.

## Figures and Tables

**Figure 1 ijerph-18-11990-f001:**
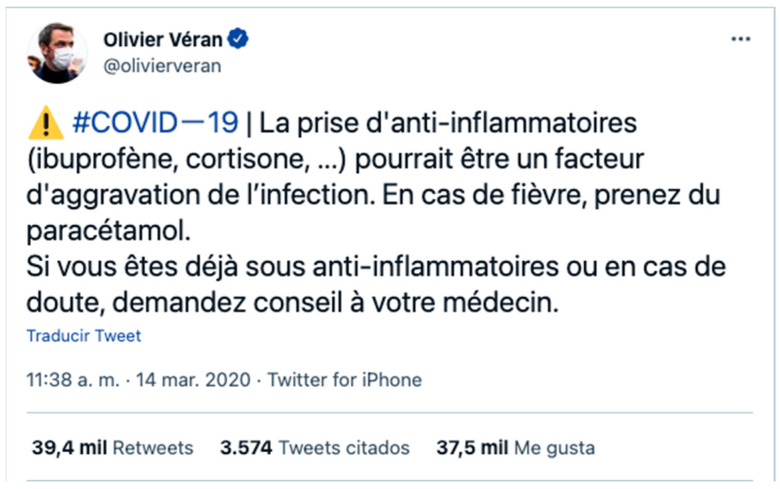
Tweet from the French Health Minister. Source: Twitter @olivierveran.

**Figure 2 ijerph-18-11990-f002:**
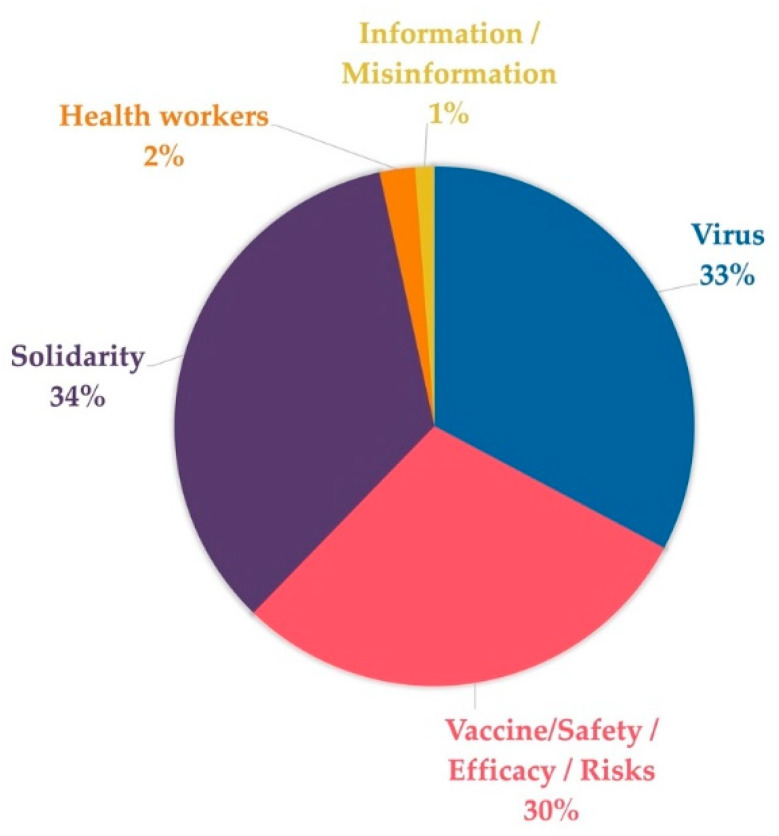
Weight (%) of the major themes in the messages.

**Figure 3 ijerph-18-11990-f003:**
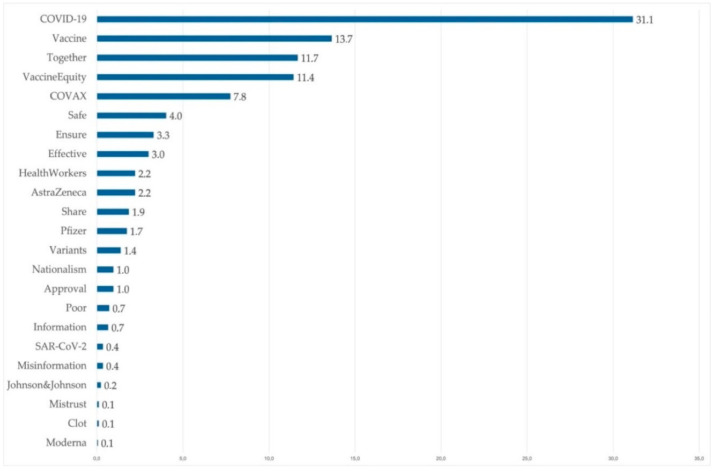
Weight (%) of the main terms included in the messages.

**Figure 4 ijerph-18-11990-f004:**
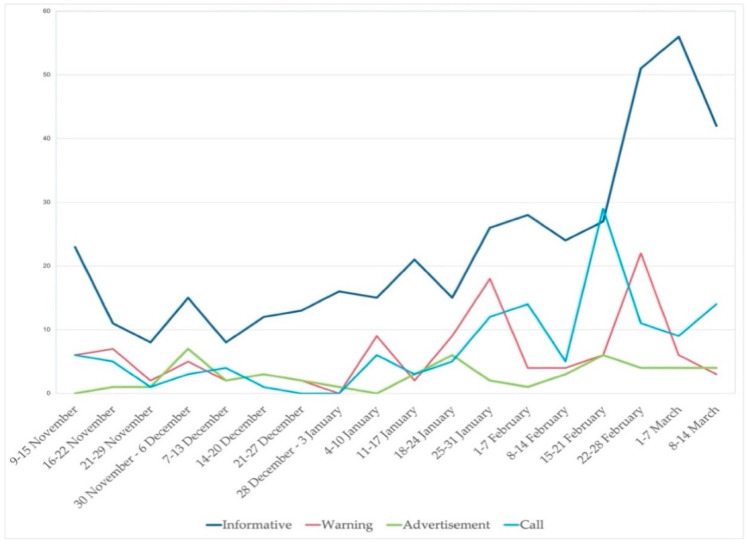
Evolution of the nature of the messages.

**Figure 5 ijerph-18-11990-f005:**
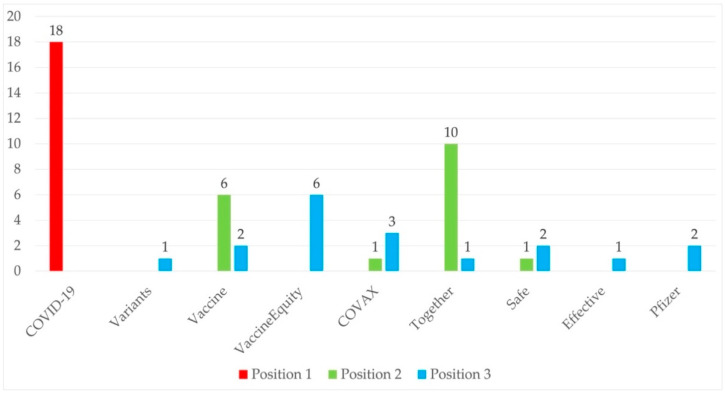
Number of weeks in which each term was among the three most frequently used.

**Figure 6 ijerph-18-11990-f006:**
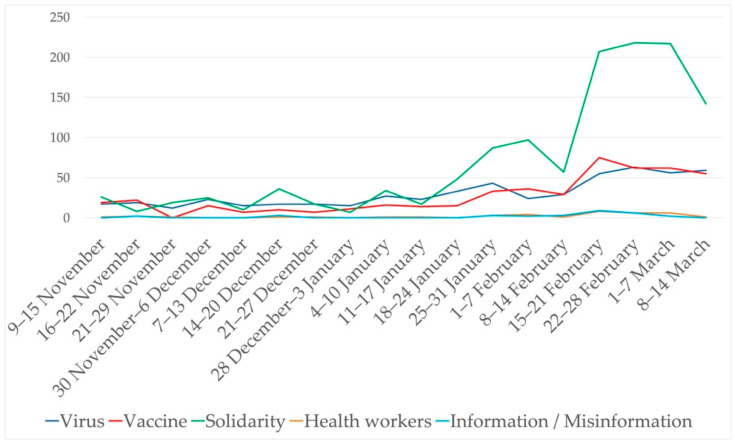
Evolution of themes in the messages.

**Table 1 ijerph-18-11990-t001:** Tweet categorization.

Criterion	Type	Description
Author	Own	An original message drafted by the organization.
	Retweet	A message that literally reproduces another message from a different account.
Reference	Direct	A message that refers explicitly to COVID-19 vaccination.
	Indirect	A message that contains no explicit reference to SARS-CoV-2 vaccines but is related to a thread on that subject.
Nature	Informative	A message that provides specific information on some aspect of the vaccines.
	Warning	A message that warns about a risk or fake news.
	Announcement	A message that promotes an activity related to the object of study.
	Appeal	A message that makes a specific request to society in general or to certain groups in particular.

## Data Availability

Publicly available datasets were analyzed in this study. This data can be found here: https://tinyurl.com/22cpmk2z.

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
