# Peer review of "The Role of Twitter in the WHO’s Fight against the Infodemic"

_ijerph, 2021, doi:10.3390/ijerph182211990_

Round 1
Reviewer 1 Report
It would be interesting to have analyzed other networks of the World Health Organization (WHO) Perhaps that is one of the limitations (only analysis of one network.
The main question of this research is directly related to the title of the paper. This question “what role does Twitter play in the fight of the World Health Organization against the infodemic generated around COVID-19?” is the central axis of the research and it is complemented and enriched with other questions mentioned in the paper, in order to get the answers that are brought out in the discussion of results and in the presentation of conclusions.
The article tackles a topic of huge relevance at the moment: the management of information on the expansion of SARS-CoV-2 and the measures put in place to fight against it. It can be said that the pandemic caused by COVID-19 has been the first health, social and economic crisis of this magnitude that has been discussed both in the real and in the virtual sphere. This situation has been a litmus test for all institutions and has made possible to measure their capacity to become authentic references and sources of reliable information for citizens. This is why the research presented in this paper stands out both because of its validity and its approach, focused on the use of Twitter by the main health organization in the world and, in particular, on the action of the WHO to fight against the fake news that have arisen about COVID-19 vaccines.
The study of the messages that the WHO posted on its official Twitter account during the period under review turned out to present relevant results. The high number of messages that make up the sample and the structure proposed for its analysis have allowed to obtain truly representative results of this subject. Furthermore, the choice of the period under review is correct because it coincides with the arrival of the first vaccines and the beginning of vaccination in different countries. Therefore, this period and the detailed study of such a large sample make this research particularly interesting due to the answers it offers and the possibilities it opens up for future research.
The methodology used in this research responds to scientific criteria and can be taken as a basis for future research. The explanation of this methodology is clear and responds to the scientific criteria that are required in a study of these characteristics. In addition, the research is based on an updated and meaning theoretical corpus that supports the analysis and allows to collect previous theories on the subject and update or exemplify them with the results of the study carried out.
The conclusions are closely related both to the arguments mentioned in the discussion of the obtained data and to the methodology and objectives used to carry out this study. These findings are in line with the research design and present new ideas for future research.
The theoretical framework of the research is made up, for the most part, of references from the last five years. This allows to give more relevance to the research and to make updated approaches from the point of view of the design of the research, the presentation and discussion of results, and the formulation of conclusions.
I believe that both the tables and the figures presented in the paper allow a visual presentation of the research data that makes easier the understanding of the results and contributes to a better understanding of the subject of study. In addition, the balance between text and visual resources makes more comfortable the reading of the paper without losing scientific rigor.
Author Response
Dear reviewer,
First of all, we would like to thank you for the comments you made in your assessment
and for the time you took to review this document. The authors thank you for the
valuable indications they will take into account to continue their line of research.
Kinds regards
Authors paper ijerph-1417788
Reviewer 2 Report
The manuscript presents a study of Tweets published by the WHO Twitter account between November 9th 2020 - March 14th 2021. The work gives an instructive overview of the WHO's messaging during an important phase of the Covid-19 pandemic. In general, I found the work is competently written. My main concern was that the novelty of the work was not sufficiently well identified or expressed. Without a stronger statement of novelty, I would not be willing to recommend publication.
The work presents an 'initial' hypothesis (lines 155-159) which was somewhat vague and open. Especially with regard to the extent that the WHO was able to combat misinformation. I don't see how the data can be used to measure the extent to which the WHO were able to combat misinformation.
Later, on line 365, the authors claim that the results 'confirm the initial hypothesis'. I don't see any evidence that the WHO have combatted any misinformation. There is likely to be a strong pro-vaccine bias among followers of the WHO Twitter account, and its messages are unlikely to spread into the communities which post misinformation. I would simply remove this claim.
Rather than presenting the 'initial hypothesis', the paper would be improved if the authors were able to more clearly identify and articulate more specific Research Questions which are addressed by the data they present in the study. However, to do this would need a significant rewrite to the Introduction to provide the right background for those Research Questions.
Author Response
Dear reviewer,
First of all, we would like to thank you for the comments you made in your assessment
and for the time you took to review this paper. The authors have responded to these
indications. We attach the answer to each of the suggestions in the table attached to
this letter.
Kinds regards
Authors paper ijerph-1417788

Reviewer 3 Report
The authors analyze the Twitter activity of WHO during COVID-19 with a particular focus on vaccines. While the topic is important, the paper currently lacks a theoretical framework to guide its research questions / hypotheses and to situate its findings in. There’s a rather superficial discussion of infodemics, but almost no discussion of the role of information during crises, why it matters, where do people get their information from, and what’s the particular role of social media in it. Similarly, the literature around misinformation in general, and on vaccines in particular is inadequate. In the lack of such theoretical guidelines, the result is a rather descriptive analysis that lacks additional depth. The analysis does not connect to any existing knowledge on epidemics and media and/or crisis communication / risk communication theories and prior findings, and thus result in a rather disconnected output that doesn’t push any knowledge or theories forward. I did my best to guide the authors towards some useful perspectives and readings for further iterations of their work. But at the moment, with the inadequate theoretical grounding and the somewhat simplistic analysis (e.g., top words), I think the paper is not ready for publication in a high impact journal.
- Despite what Hao and Basu said, COVID-19 was NOT the first epidemic to be experienced via social networks. Zika (2016), Ebola (2014), and Swine flu (2009) occurred at a time when many got their news online, to name just a few.
- line 34 – the first time you use the term WHO you need to provide the full name
- line 41 – don’t use the term “fake news” – what you mean here is misinformation
- In the same context, there are many specific examples for misinformation during epidemic. See for example Bode & Vraga’s work on Zika (http://dx.doi.org/10.1080/10410236.2017.1331312)
- Similarly, I think you should expend quite a lot on the role of information during public health crises and epidemics, specifically. What information people need? Where do they look it up? What do they get from the news media? (see information needs in the CDC’s CERC framework for example: https://emergency.cdc.gov/cerc/resources/pdf/cerc_2014edition.pdf ; sense making in Weick’s work http://onlinelibrary.wiley.com/doi/10.1111/j.1467-6486.1988.tb00039.x/abstract)
- Similarly, you need to talk more in detail about misinformation, its spread online, and why it’s often resistant to change. There’s such a huge body of literature you can take from for this, but a good starting point would be Lewandowsky 2012 (http://journals.sagepub.com/doi/10.1177/1529100612451018)
- Within this missing discussion, a particular part should be dedicated to vaccine misinformation in general (http://www.sciencedirect.com/science/article/pii/S0264410X09019264) , misinformation about novel vaccines (https://academic.oup.com/jpubhealth/article/40/4/e531/4925540 ) and COVID-19 vaccine misinformation specifically (http://www.sciencedirect.com/science/article/pii/S027795362030575X)
- When referring to mis and disinformation (e.g., line 61) define what they are and what’s the difference between the two. There are good definitions here in Freelon et al (https://doi.org/10.1080/10584609.2020.1723755) and an example for disinformation around vaccines specifically could be found in Walter et al in regard to the Russian interference in American Twitter (https://ajph.aphapublications.org/doi/10.2105/AJPH.2019.305564 )
- line 78 – not only to track public opinion but also to facilitate two-way communication during crises (http://www.sciencedirect.com/science/article/pii/S0196655316306198)
- line 110 – what do you mean “their study shows the importance of information transparency” – how?
- not clear what guidelines you’re talking about in line 136. be more specific…
- How can you determine the intent of terms by looking at text alone (line 144) – that sounds like a task that requires interviewing those who wrote them. I can’t see how top words could reveal intentions
- The “hypothesis” in line 155 is not really an hypothesis, but an assumption
- If one of your goals is to connect to topics in the news (line 171), you need to discuss studies on the coverage of epidemics in and COVID-19 specifically (https://www.cambridge.org/core/product/identifier/S0008423920000396/type/journal_article)
- I’m not sure I follow the logic of the collection procedure – if you used the word “vaccine” in your search term, why did you need to filter messages not about vaccines (line 177)
- What theoretical constructs or models were used to build the codebook in Table 1?
- As I said earlier, I really dislike the term “intention” in your analysis – you don’t know what their intentions were. You know what the content of their tweets was
- How were the topics identified (line 250)? Manually? What was the process?
Author Response

(The authors gave the same response as above.)

Round 2
Reviewer 2 Report
Thank you for addressing my concerns.
The paper is now largely appropriate for publication. I would recommend that the authors carefully proof read the paper. Some of the English does not look grammatically correct to me.
Author Response
Dear reviewer,
First of all, we would like to thank you for the comments you made in your assessment
and for the time you took to review this document. The authors thank you for the
valuable indications they will consider continuing their line of research.
Kinds regards
Authors paper ijerph-1417788

Reviewer 3 Report
The authors made some changes, including improving the breadth of their literature review. However, I’m still not convinced by the method, novelty or contribution of the work. The authors write in line 149 that “the main innovation of this research is the approach taken” but don’t explain what this approach is or why it is novel. As far as I can see, there is still no organized theoretical framework to guide the content analysis. You mention that you follow, in part, the likes of Strauss and Corbin, but don’t explain what they proposed. For example, the categorization of information/warning/announcement/appeal – what is it based on? What’s the definition of each of these? How did your codebook define and operationalize each? The manuscript is still very lacking in terms of description of methods. You say that “the research has made it possible to define topics” – I don’t understand what you mean by that. How were topics defined? What was the content analysis based on? How did you come up with these labels? How did you validate it? So while you did improve the literature a bit (it's still not very thorough) you did not address my main concerns with the paper. It remains largely descriptive and not well connected to any particular theoretical line of work.
Finally, I understand and acknowledge the challenges of writing in a second language, but I do believe the manuscript requires a professional edit from a native English speaker. Many phrases feel unnatural, making it harder to read and comprehend.
Minor comments: in the abstract- your results cannot show that WHO are “decidedly committed” -what you can say is that it shows the WHO have used Twitter for… In addition, you still use the term fake news in multiple places.
Author Response
Dear reviewer,
First of all, we would like to thank you for the time you took to review this paper. The
authors have responded to these indications. We attach the answer to each of the
suggestions in the table attached to this letter.
Kinds regards
Authors paper ijerph-1417788